# Melatonin Improves Fatty Liver Syndrome by Inhibiting the Lipogenesis Pathway in Hamsters with High-Fat Diet-Induced Hyperlipidemia

**DOI:** 10.3390/nu11040748

**Published:** 2019-03-30

**Authors:** Tzu-Hsuan Ou, Yu-Tang Tung, Ting-Hsuan Yang, Yi-Wen Chien

**Affiliations:** 1School of Nutrition and Health Sciences, Taipei Medical University, Taipei, Taiwan; reira427@gmail.com (T.-H.O.); vicky821018@yahoo.com.tw (T.-H.Y.); 2Graduate Institute of Metabolism and Obesity Science, Taipei Medical University, Taipei, Taiwan; f91625059@tmu.edu.tw; 3Nutrition Research Center, Taipei Medical University Hospital, Taipei, Taiwan

**Keywords:** melatonin, hamsters, dyslipidemia, diet-induced obesity, lipogenic enzymes

## Abstract

The aim of this study was to investigate the effect of melatonin on hepatic lipid metabolism in hamsters with high-fat diet (HFD)-induced dyslipidemia. Male Syrian hamsters were kept on either a chow control (C) or HFD for four weeks. After four weeks, animals fed the HFD were further randomly assigned to four groups: high-fat only (P), melatonin low-dosage (L), medium-dosage (M), and high-dosage (H) groups. The L, M, and H groups, respectively, received 10, 20, and 50 mg/kg/day of a melatonin solution, while the P and C groups received the ethanol vehicle. After eight weeks of the intervention, results showed that a low dose of melatonin significantly reduced HFD-induced hepatic cholesterol and triglycerides; decreased plasma cholesterol, triglycerides, and low-density lipoprotein cholesterol; and increased plasma high-density lipoprotein cholesterol (*p* < 0.05). In addition, melatonin markedly decreased activities of the hepatic lipogenic enzymes, acetyl-CoA carboxylase (ACC) and fatty acid synthase (FAS) (*p* < 0.05), and elevated the relative hepatic carnitine palmitoyltransferase-1α expression in hamsters with HFD-induced hyperlipidemia. Consequently, melatonin reduced activities of the hepatic lipogenic enzymes, ACC and FAS. In summary, chronic melatonin administration improved HFD-induced dyslipidemia and hepatic lipid accumulation in Syrian hamsters with HFD-induced dyslipidemia, which might have occurred through inhibiting the lipogenesis pathway.

## 1. Introduction

There is a worldwide epidemic of obesity—a metabolic disorder associated with many deadly diseases and related comorbidities, such as dyslipidemia, type II diabetes, and cardiovascular disease [1]. The prevalence of obesity is increasing worldwide, and, if such trends continue, by 2030 up to 57.8% of the world’s adult population (3.3 billion people) could be either overweight or obese [2]. Therefore, effective methods based on newly identified and modifiable risk factors are indispensable for combating obesity.

Melatonin (*N*-acetyl-5-methoxytryptamine) is a neurohormone in vertebrates mainly produced by the pineal gland, from the essential amino acid tryptophan. Melatonin regulates various physiological functions, including circadian rhythms, thermoregulation, immune responses, and seasonal reproductive functions [3]. Furthermore, as a free radical scavenger and mitochondrion-targeting antioxidant, melatonin has a strong antioxidant capacity against oxidative stress [4,5]. Several metabolites of melatonin also act as antioxidants [6,7,8]. Previous studies showed that melatonin can affect body mass, adiposity, and both energy intake and expenditure [9]. In a pinealectomized animal model, prediabetic symptoms of glucose intolerance and insulin resistance (IR) in hepatic, adipose, and skeletal muscle tissues were discovered [10,11]. Prunet-Marcassus et al. [12] also demonstrated that melatonin could regulate the body mass (BM) gain in a diet-induced obesity model, improve glucose homeostasis, and prevent the dysregulation of lipid metabolism.

Although previous studies have investigated the effects of melatonin on lipid regulation in rodents, there has been less focus on the regulation of the hepatic fatty acid de novo pathway and fatty acid β-oxidation in diet-induced obesity (DIO) hamster models [13]. The aim of this study was to investigate the effect of melatonin on hepatic lipid metabolism in hamsters with high-fat diet (HFD)-induced dyslipidemia, in which blood lipid profiles respond to diets in a predictive manner similar to that in humans.

## 2. Materials and Methods

### 2.1. Animals and the Experimental Design

Forty male Syrian hamsters (*Mesocricetus auratus*) were obtained at an age of 4 weeks from the National Laboratory Animal Center (NLAC, Taipei, Taiwan). All animals were housed two or three per plastic cage and maintained under standard conditions with a 12-h light/dark cycle (lights on at 08:00) and at a temperature of 22 ± 2 °C. After the acclimatization period, hamsters were kept on either a standard chow diet (C group) or a HFD for 4 weeks. Compositions of the standard chow diet and HFD are given in Table 1. To confirm that the dyslipidemia model was established, hamster BM and plasma lipid profiles were measured at the end of the fourth week. After 4 weeks, hamsters fed the HFD were further randomly assigned into four groups, including a high-fat only (P) group, and melatonin low-dosage (L), medium-dosage (M), and high-dosage (H) groups. Melatonin (Sigma-Aldrich, St Louis, MO, USA) was dissolved in 6% ethanol and given by gavage. For the L, M, and H groups, melatonin concentrations of 10, 20, and 50 mg/mL were prepared, respectively. The hamsters were administrated with a volume of 1 mL of melatonin solution/kg of BM. Therefore, the L, M, and H groups received 10, 20, and 50 mg/kg/day of a melatonin solution, respectively, at 18:00, while the P and C groups received the 6% ethanol vehicle. After 8 weeks of the intervention, animals were anesthetized with an intramuscular injection of a Zoletil–Rompun mixture after overnight fasting. Blood samples were collected into EDTA vacutainer tubes by heart puncture and, after being centrifuged for 10 min (1500× *g*) at 4 °C, were stored at −80 °C. Inguinal, epididymal, and retroperitoneal white adipose tissues were dissected and immediately weighed. All procedures were approved by the Institutional Animal Care and Use Committee of Taipei Medical University (IACUC number: LAC-2017-0192).

### 2.2. Biochemical Parameter Measurements

Plasma glucose, total cholesterol (TC), triglycerides (TG), and high-density lipoprotein cholesterol (HDL-C) levels were determined colorimetrically using commercial kits (Randox Laboratories, Antrim, UK). Low-density lipoprotein cholesterol (LDL-C) was calculated according to the equation: LDL-C = TC − HDL-C − TG/5. The plasma insulin level was determined using a Mercodia Ultrasensitive Rat Insulin ELISA (enzyme-linked immunosorbent assay) kit (Mercodia AB, Uppsala, Sweden). Total liver lipids were extracted via the method reported by Folch et al. [14], and hepatic TC and TG contents were quantified using commercial kits (Randox Laboratories).

### 2.3. Real-Time Reverse-Transcription Polymerase Chain Reaction (RT-PCR)

Total RNA was extracted from liver tissue using the TRI Reagent (Sigma-Aldrich, Steinheim, Germany) according to the manufacturer’s instructions, and 3 µg of total RNA was reverse-transcribed with a RevertAid First Strand cDNA Synthesis Kit (Thermo Scientific, Thermo Scientific, Waltham, MA, USA). A real-time quantitative (q)PCR was performed according to SYBR Green/ROX qPCR Master Mix (2X) (Thermo Scientific) in a 25 μL total reaction volume using the 7300 Real-Time PCR System (Applied Biosystems, Foster City, CA, USA). To evaluate gene expressions, real-time RT-PCRs were performed for five genes (*FAS*, *ACC*, *SREBP-1c*, *PPAR-α*, and *CPT1α*) using complementary (c) DNA from liver tissue. The β-actin gene was used as an internal control.

### 2.4. Hepatic Fatty Acid Metabolic Enzyme Measurements

The protein concentration of each sample was determined using a Bio-Rad Protein assay kit (Bio-Rad Laboratories, Hercules, CA, USA). Fatty acid synthase (FAS) activity was measured as the malonyl coenzyme A-dependent oxidation of nicotinamide adenosine dinucleotide phosphate (NADP) in the presence of acetyl coenzyme A. acetyl-CoA carboxylase (ACC) was determined as described by Numa et al. [15]. Briefly, approximately 200 μg of unfrozen liver tissue was homogenized in 2 mL of 0.3 M mannitol, 100 mM 4-(2-hydroxyethyl)-1-piperazineethanesulfonic acid (HEPES), and 1 mM ethylene glycol tetra-acetic acid (EGTA) buffer (pH 7.2), and then centrifuged at 12,000× *g* for 10 min at 4 °C. The resulting post-nuclear supernatant (PNS) was analyzed for activities of the hepatic enzymes FAS and ACC. The reaction was run for 5 min at 37  °C, and enzyme activities were expressed as nmol/min/mg liver protein.

### 2.5. Histologic Analysis

Liver tissue was fixed in 10% buffered formaldehyde. Before proceeding to the next step, tissue was soaked in absolute ethanol overnight and embedded in paraffin. Sections were then stained with hematoxylin and eosin (H&E).

### 2.6. Statistical Analysis

All results are presented as the mean ± standard error of the mean (SEM). Changes in BMs, glucose levels, and lipid profiles between group C and the HFD groups in the first 4 weeks were analyzed by Student’s *t*-test. To evaluate differences between groups during the 8-week intervention period, a two-way ANOVA and Duncan’s multiple-range tests were used in SPSS vers. 22 (SPSS, IBM, Chicago, IL, USA). Differences were considered statistically significant at *p* < 0.05.

## 3. Results

### 3.1. Effects of Melatonin on BM and Tissue Weights

After four weeks of HFD treatment, the HFD group had gained significantly more weight than the C group (*p* < 0.05). Plasma fasting glucose (114.9 ± 3.7 mg/dL), TC (184.5 ± 5.2 mg/dL), TG (142.5 ± 10.5 mg/dL), and LDL-C (121.1 ± 4.5 mg/dL) levels in the HFD group were also significantly higher than those in the C group (fasting glucose: 87.8 ± 5.5 mg/dL, *p* < 0.05; TC: 105.4 ± 9.6 mg/dL, *p* < 0.05; TG: 81.0 ± 6.2 mg/dL, *p* < 0.05; LDL-C: 52.9 ± 9.1 mg/dL, *p* < 0.05). Thus, the hyperlipidemic model had clearly been established. At the end of the eight-week intervention, there were no significant differences in food intake, kidney weight, or inguinal, epididymal, or retroperitoneal fat tissues between the groups, but the total weight gain and energy intake in the P group were significantly greater than those in the C group (Table 2). The liver weight and relative liver weight were significantly higher in the P group than in the C group, and these were significantly ameliorated after melatonin supplementation (Table 2).

### 3.2. Effects of Melatonin on Blood and Hepatic Biochemical Parameters

To determine the effects of melatonin on lipid metabolism, plasma and hepatic lipid levels were measured. After the eight-week intervention, plasma TC, LDL-C (Figure 1), and hepatic TG (Figure 2) levels had significantly increased in the P group and were significantly lower after melatonin supplementation (*p* < 0.05). The plasma TG level was also higher in the P group and was only significantly alleviated in the L group (Figure 1; *p* < 0.05). Moreover, although there was no difference in plasma HDL-C levels between the C and P groups, there were significant increases in the L and M groups after eight weeks of melatonin supplementation (Figure 1; *p* < 0.05). The hepatic TC level had significantly increased in the P group and was significantly lower only in the L and M groups (Figure 2; *p* < 0.05). In addition, the level of plasma insulin was examined, but there were no significant differences between groups (Figure 1). H&E staining of liver tissue biopsies showed that all HFD treatment groups developed steatosis and microvesicular fatty changes with slight Kupffer cell hyperplasia. The melatonin intervention diminished the extent of steatosis (Figure 3).

### 3.3. Effects of Melatonin on Hepatic mRNA Expressions and Enzyme Activities

SREBP-1c is a transcriptional factor that regulates the expressions of the ACC and FAS enzymes that are involved in fatty acid synthesis. Unexpectedly, relative mRNA expressions of *FAS* and *ACC* did not significantly differ between all groups (Figure 4). Compared to the C group, *SREBP-1c* expression was lower in almost all HFD treatment groups except the L group (Figure 4). However, melatonin significantly attenuated the HFD-induced elevated ACC and FAS enzyme activities (Figure 5; *p* < 0.05). Furthermore, mRNA expressions of *PPAR-α* and *CPT-I* were examined, and results showed that melatonin significantly alleviated the HFD-induced decrease in the relative *CPT-1* mRNA expressions in the M group (Figure 4; *p* < 0.05). Relative *PPAR-α* mRNA expressions among all HFD groups did not statistically significantly differ, but were lower than those of the C group (Figure 4).

In the present study, we established a hyperlipidemic model to examine the effects of melatonin on impaired lipid metabolism, and results showed that hamsters fed an HFD for four weeks certainly exhibited significantly elevated BMs compared to those fed a normal diet. We further administrated doses of 10, 20, or 50 mg/kg of melatonin to the obese hamsters and, after the eight-week intervention, the final BMs and adipose tissues, including inguinal, epididymal, and retroperitoneal fatty tissues, had not changed among all groups. Wongchitrat et al. [16] found that a single dose of 10 mg/kg melatonin for 6 weeks via an intraperitoneal route to DIO rats had no significant effect on the BM, body fat, or epididymal and retroperitoneal adipose tissue weights. Prunet-Marcassus et al. [12] also suggested that rats concurrently treated with a single dose of 30 mg/kg melatonin and an HFD tended to have a lower body fat mass, but the decrease was not obvious. Our findings agree with those of previous studies [12,16]. A previous study suggested that melatonin was effective in BM regulation [17]—DIO rats receiving melatonin (4 mg/kg/day) in drinking water for 14 weeks had significantly reduced BMs compared to the untreated group, but the abdominal fat weight had not changed [17]. However, continuous melatonin administration is not likely to be suitable for humans since sleepiness can be one of the obvious side effects at higher dosages [18]. The difference between the HFD and HFD with melatonin might have become statistically significant with a longer treatment period and a different route of melatonin administration because we gave melatonin only once a day before nighttime, rather than continuously provided in drinking water.

In our DIO model, hamsters fed the HFD for four weeks had significantly higher plasma TC, TG, and LDL-C levels. After the eight-week intervention period, results indicated that daily melatonin administration significantly improved the impaired dyslipidemic profile, and that a low dose of 10 mg/kg of melatonin was sufficient to exert this effect [19,20,21,22,23]. Several studies revealed that melatonin can be useful in lowering cholesterol and TG concentrations. Kitagawa et al. [19] administered 10 mg/kg of melatonin to obese rats with metabolic syndrome, and found it could reduce serum TC, TG, and LDL-C levels, and increase the serum HDL-C level. Several mechanisms might explain the hypolipidemic effects of melatonin. Previous studies have suggested that melatonin can lower cholesterol absorbed in the intestines [20], increase catabolism of cholesterol by bile acids, and inhibit cholesterol biosynthesis [21] and interactions with LDL-C receptors [22]. The results of this study also demonstrated that 10 mg/kg melatonin reduced liver TC and TG levels. The protective effect might be associated with melatonin and its metabolites, which have antioxidant activities in the liver. According to Pan et al. [23], 5 and 10 mg/kg melatonin caused reductions in liver TC and TG contents, as well as the level of malondialdehyde (MDA), a liver lipid peroxidation parameter. Furthermore, liver superoxide dismutase (SOD) and glutathione peroxidase (GSH-Px) levels were elevated in their DIO rat model.

Dysregulation of lipid metabolism in the liver—especially hepatic de novo fatty acid synthesis—could be attributed to the development of non-alcoholic fatty liver disease (NAFLD) [24]. ACC catalyzes malonyl-CoA synthesis, the rate-limiting step in fatty acid biosynthesis, whereas FAS catalyzes the final step in fatty acid synthesis. Chen et al. [25] indicated that lipopolysaccharide (LPS)-induced upregulation of hepatic *FAS* and *ACC* mRNA levels were significantly attenuated in hamsters after they were treated with melatonin. SREBP-1c, an important transcription factor that regulates expressions of enzymes in fatty acid synthesis, was also attenuated after melatonin administration in LPS-treated mice [25]. Unexpectedly, relative mRNA expressions of *FAS*, *ACC*, and *SREBP-1c* were not elevated after HFD treatment in the present study, inconsistent with the results of enzyme activities. The plasma and hepatic lipid profiles were significantly elevated in our hyperlipidemic hamster model, and liver biopsy staining with H&E showed fatty changes in hepatocytes. One possible explanation is that the HFD cholesterol in our study mainly caused microvesicular instead of macrovesicular fatty changes, and did not produce IR. The common form of fatty liver is characterized by a mixture of micro- and macrovesicular steatosis [26], and several studies demonstrated that IR clearly develops in NAFLD and other metabolic diseases [27]. Nevertheless, Kainuma et al. [28] established a cholesterol-fed model with more predominant microvesicular steatosis, which showed no obesity or IR, but increased concentrations of lipid peroxide (LPO) in plasma and liver tissues. They suggested that this kind of fatty liver disease might be primarily caused by impairment of mitochondrial β-oxidation [24]. Our findings agree with previous studies in which the relative mRNA expression of *CPT-I*, which catalyzes the crucial step of mitochondrial β-oxidation, decreased after the HFD intervention, as expected. Furthermore, we found that a dose of 20 mg/kg melatonin significantly enhanced *CPT-I* mRNA expression. Few studies have investigated the effect of melatonin on the liver mitochondrial β-oxidation pathway. Heo et al. [29] pointed out that the level of *CPT-I* did not significantly change after HFD or melatonin administration. Based on these previous works, we considered that melatonin might participate in regulating hepatic fatty acid β-oxidation and ameliorate lipid accumulation. The mechanism by which melatonin modulates fatty liver syndrome is proposed in Figure 6.

In conclusion, chronic melatonin administration could improve HFD-induced dyslipidemia and hepatic lipid accumulation in DIO Syrian hamsters, which occurred through regulating lipogenic enzymes and lipid oxidation at the mRNA level.

## Figures and Tables

**Figure 1 nutrients-11-00748-f001:**
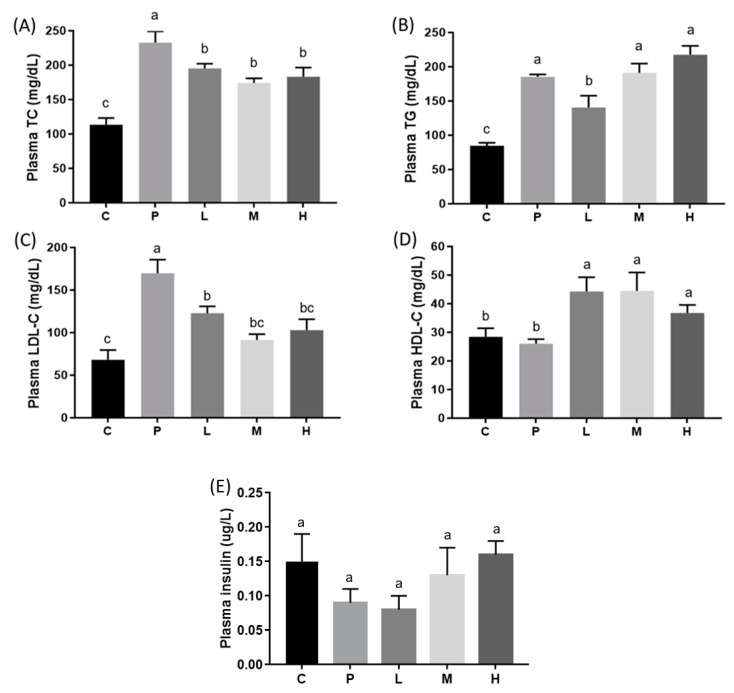
Plasma (**A**) total cholesterol (TC), (**B**) triglyceride (TG), (**C**) low-density lipoprotein cholesterol (LDL-C), (**D**) high-density lipoprotein cholesterol (HDL-C), and (**E**) insulin concentrations in hamsters fed a high-fat diet (HFD) after a melatonin intervention for eight weeks. Daily gavage administration of melatonin at 10 (L), 20 (M), or 50 (H) mg/kg, or the ethanol vehicle to hamsters fed an HFD (P) or control diet (C). All values are the mean ± SEM (*n* = 6–8). Different letters (a, b, c) indicate a significant difference at *p* < 0.05 determined using two-way ANOVA.

**Figure 2 nutrients-11-00748-f002:**
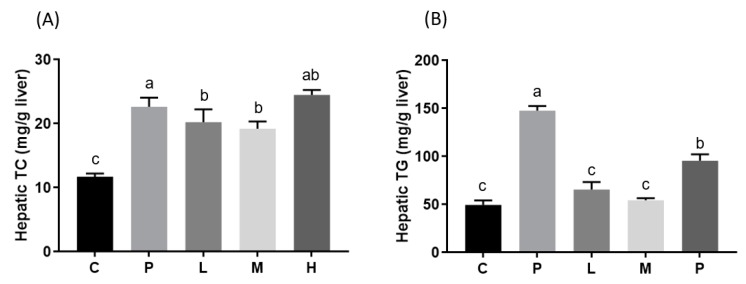
Hepatic (**A**) TC and (**B**) TG concentrations in hamsters fed a HFD after a melatonin intervention for eight weeks. Daily gavage administration of melatonin at 10 (L), 20 (M), or 50 (H) mg/kg, or the ethanol vehicle to hamsters fed a HFD (P) or control diet (C). All values are the mean ± SEM (*n* = 6–8). Different letters (a, b, c) indicate a significant difference at *p* < 0.05 determined using two-way ANOVA.

**Figure 3 nutrients-11-00748-f003:**
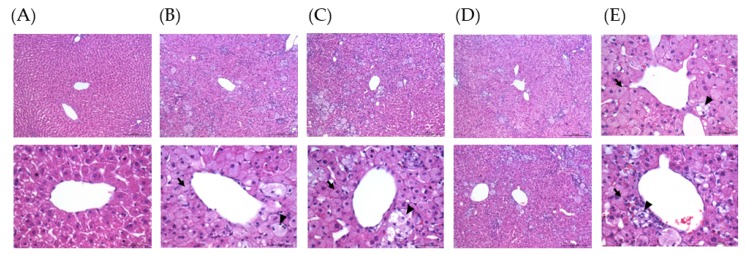
Hepatic histopathology of hamsters fed an HFD after a melatonin (MEL) intervention for eight weeks (200× magnification): (**A**) Control with a regular diet, (**B**) HFD without MEL), (**C**) HFD with 10 mg/kg MEL, (**D**) HFD with 20 mg/kg MEL, and (**E**) HFD with 50 mg/kg MEL. Microvesicular fatty changes (arrow) were observed. Slight Kupffer cell hyperplasia was observed (arrowheads).

**Figure 4 nutrients-11-00748-f004:**
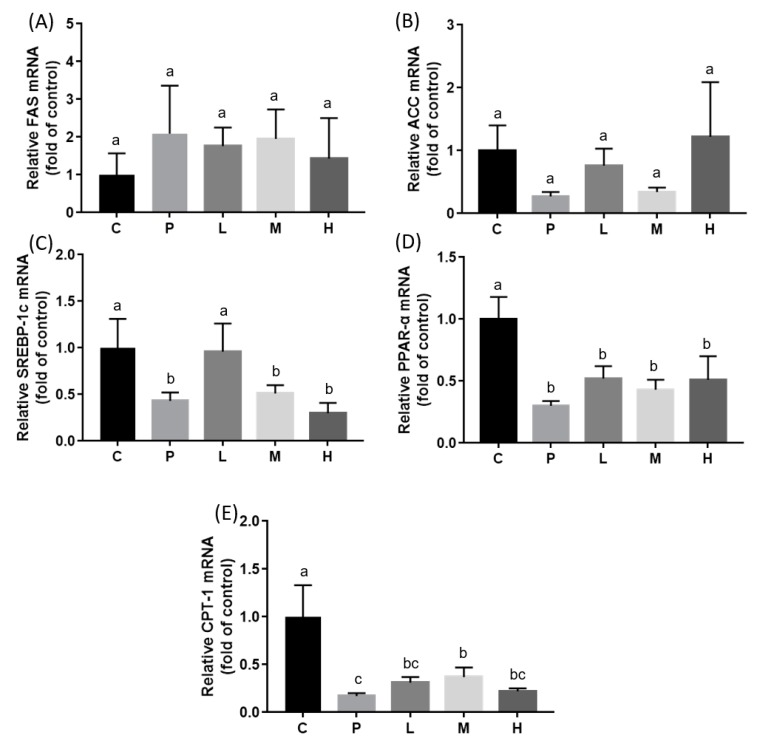
Hepatic mRNA expressions of (**A**) fatty acid synthase (FAS), (**B**) acetyl-CoA carboxylase (ACC), (**C**) sterol regulatory element-binding protein (SREBP)-1c, (**D**) peroxisome proliferator-activated receptor (PPAR)-α, and (**E**) carnitine palmitoyltransferase I (CPT)-I in hamsters fed an HFD after a melatonin intervention for eight weeks. Daily gavage administration of melatonin at 10 (L), 20 (M), or 50 (H) mg/kg, or the ethanol vehicle to hamsters fed an HFD (P) or control diet (C). All values are the mean ± SEM (*n* = 6–8). Different letters (a, b, c) indicate a significant difference at *p* < 0.05 determined using two-way ANOVA.

**Figure 5 nutrients-11-00748-f005:**
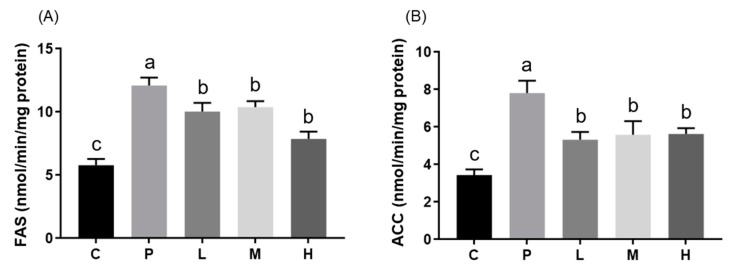
Hepatic lipogenic enzyme (**A**) ACC and (**B**) FAS activities in hamsters fed a HFD after a melatonin intervention for eight weeks. Daily gavage administration of melatonin at 10 (L), 20 (M), or 50 (H) mg/kg, or the ethanol vehicle to hamsters fed an HFD (P) or control diet (C). All values are the mean ± SEM (*n* = 6–8). Different letters (a, b, c) indicate a significant difference at *p* < 0.05 determined using two-way ANOVA.

**Figure 6 nutrients-11-00748-f006:**
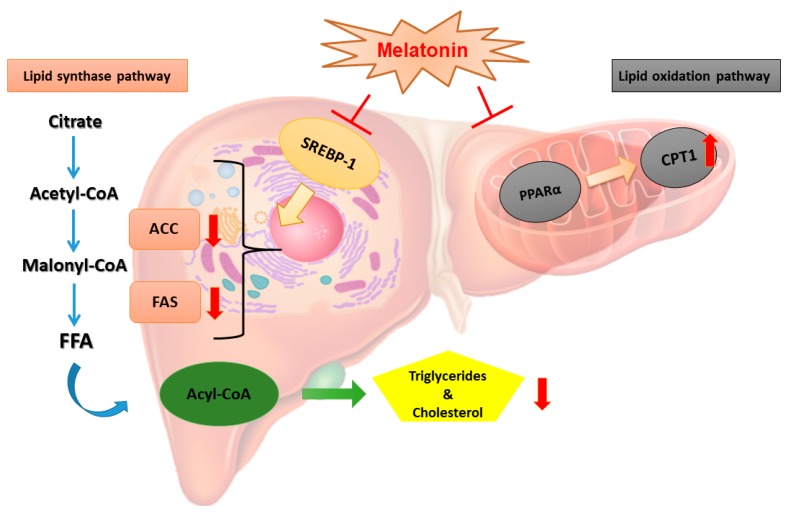
Proposed mechanism of melatonin-mediated modulation of the lipid metabolism pathway in hamsters fed an HFD. The effects of melatonin on the fatty liver of hamsters fed an HFD may occur through inhibition of metabolic enzymes of the lipogenic pathway and lipid oxidative pathway. The diagram demonstrates that melatonin might inhibit the lipogenic enzymes ACC and FAS during cholesterol synthesis.

**Table 1 nutrients-11-00748-t001:** Ingredient composition of the experimental diets fed to hamsters.

(g/kg Diet)	Control Diet	High-Fat Diet
Casein	140	140
L-Cysteine	1.8	1.8
Corn starch	620.7	558.7
Sucrose	100	100
Cellulose	50	50
Soybean oil	40	40
Lard	0	110
AIN-93 mineral mix	35	35
AIN-93 vitamin mix	10	10
Choline bitartrate	2.5	2.5
Cholesterol	0	2
Calories (kcal/kg diet)	3850	4592

**Table 2 nutrients-11-00748-t002:** Body mass (BM), food intake, energy intake, feed efficiency, organ weights, and fat mass after eight weeks of the melatonin intervention.

	C	P	L	M	H
Initial BM (g)	137.0 ± 3.1	142.2 ± 4.3	135.8 ± 3.7	142.8 ± 5.6	144.1 ± 6.0
Final BM (g)	141.2 ± 1.7 ^b^	159.4 ± 4.6 ^ab^	154.2 ± 6.3 ^ab^	153.9 ± 7.8 ^ab^	162.4 ± 9.1 ^a^
Total weight gain (g)	4.1 ± 2.3 ^b^	17.2 ± 3.5 ^a^	18.4 ± 3.5 ^a^	11.1 ± 6.8 ^ab^	18.3 ± 3.9 ^a^
Food intake (g/day)	7.7 ± 0.11	7.8 ± 0.22	7.3 ± 0.32	7.7 ± 0.15	7.3 ± 0.18
Energy intake (kcal/day)	29.8 ± 0.41 ^b^	36.0 ± 0.99 ^a^	33.7 ± 1.49 ^a^	35.5 ± 0.70 ^a^	33.7 ± 0.81 ^a^
Feed efficiency (%)	7.6 ± 4.2 ^b^	31.2 ± 6.3 ^ab^	34.8 ± 6.6 ^a^	20.5 ± 12.5 ^ab^	35.7 ± 7.6 ^a^
Liver weight (g)	3.97 ± 0.22 ^c^	8.25 ± 0.25 ^a^	6.80 ± 0.47 ^b^	7.15 ± 0.24 ^b^	7.92 ± 0.45 ^ab^
Kidney weight (g)	0.86 ± 0.02	0.92 ± 0.06	0.92 ± 0.04	0.95 ± 0.04	0.92 ± 0.03
Inguinal fat weight (g)	3.50 ± 0.40	4.46 ± 0.61	4.08 ± 0.76	4.51 ± 0.51	5.06 ± 0.64
Epididymal fat weight (g)	2.71 ± 0.14	4.13 ± 0.74	3.05 ± 0.36	3.13 ± 0.39	3.39 ± 0.35
Retroperitoneal fat weight (g)	2.10 ± 0.11	2.75 ± 0.24	2.30 ± 0.31	2.86 ± 0.22	2.57 ± 0.33
Relative liver weight (g/100 g BM)	2.82 ± 0.14 ^c^	5.20 ± 0.31 ^a^	4.35 ± 0.12 ^b^	4.49 ± 0.06 ^b^	4.90 ± 0.27 ^ab^
Relative epididymal fat weight (g/100 g BM)	1.93 ± 0.08	2.55 ± 0.41	1.94 ± 0.17	1.93 ± 0.19	2.09 ± 0.20

Daily gavage administration of melatonin at 10 (L), 20 (M), or 50 mg/kg (H), or the ethanol vehicle to hamsters fed a high-fat diet (P) or control diet (C). All values are the mean ± standard error of the mean (SEM) (*n* = 6–8). Different letters (a, b, c) in the same row indicate a significant difference at *p* < 0.05 determined using two-way ANOVA. Formula: feed efficiency = weight gain/food intake × 100%.

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
