# Peer review of "Melatonin Improves Fatty Liver Syndrome by Inhibiting the Lipogenesis Pathway in Hamsters with High-Fat Diet-Induced Hyperlipidemia"

_nutrients, 2019, doi:10.3390/nu11040748_

Reviewer 1 Report

Disorders of lipid metabolism are very important medical and social problems.  Hence, effective and novelty  treatment are searching for its improvement. The results of previous study showed that melatonin exerts a positive effects on liver function and lipid profile, but its mechanism of working is not satisfactory clarified. The aim of the reviewed study was to investigate the effect of melatonin on hepatic lipid metabolism in hamsters with high-fat diet (HFD)-induced dyslipidemia in which blood lipid profiles respond to diets in a predictive manner similar to that by humans.

The structure of work and the methods used are appropriate and well described to clarify the questions that the authors asked. Presented data were well controlled and statistical analysis is relevant. The obtained results are very important. The authors demonstrated significant influence of melatonin on lipid metabolism in liver trough inhibition of metabolic enzymes on the lipogenic pathway and lipid oxidative pathway. Especially, melatonin decreased activities of acetyl-CoA carboxylase(ACC) and fatty acid synthase (FAS). Subsequently, melatonin supplementation, even in low-dosage (10/kg/day),  reduced hepatic cholesterol and triglycerydes, decreased plasma LDL-cholesterol  and triglycerydes , and increased HDL-cholesterol.

These results have a practical implications, but next clinical studies are necessary in the future. Particularly,  it has not been determined what optimal doses of melatonin should be used.

Author Response

Thank you very much for the reviewer’s valuable comments. The paper was revised based on their constructive suggestions. The following are the point-by-point responses to the comments:

For reviewer #1:

We appreciate the reviewer affirmation. In this study, the results demonstrated significant influence of melatonin on lipid metabolism in liver trough inhibition of metabolic enzymes on the lipogenic pathway and lipid oxidative pathway. Especially, melatonin decreased activities of acetyl-CoA carboxylase(ACC) and fatty acid synthase (FAS). Subsequently, melatonin supplementation, even in low-dosage (10 mg/kg/day), reduced hepatic cholesterol and triglycerydes, decreased plasma LDL-cholesterol and triglycerydes, and increased HDL-cholesterol.

As a commenter's suggestion, we will do clinical research in the future. Concerning our results showed effective dose started at 10 mg/kg/day (equivalent to 1.36 mg/kg/d in human). I have added the point in our revised manuscript (Page 9, Lines 266-268).

Reviewer 2 Report

The experiment is extremely interesting and the results are very promising. However, some minor revision should be done.

1. Could you provide the information about the concentration of melatonin in ethanol solution  given to hamsters in the experiment?

2. Time of daily melatonin dosage is not provided - was it in the evening, in accordance with the physiological melatonin secretion? This information is very important for the practical use of the results and planning further experiments, especially in humans.

3. In Material and Methods section it should be rather adipose/liver tissue sample/samples, not adipose/liver tissues.

4. The term "body mass' (BM) instead of "body weight' (BW) should be used throughout the manuscript.

5. In Table 2, the superscripts a, b, c are not described enough - which groups are compared as a, b, or c? The  explanation is incomprehensible. The same problem is in Figures 1, 2, 4.

Author Response

Q1. Could you provide the information about the concentration of melatonin in ethanol solution given to hamsters in the experiment?

Answer: Melatonin (Sigma-Aldrich, St Louis, MO, USA) was dissolved in 6% ethanol and given by gavage. For the L, M and H groups, melatonin concentrations of 10, 20 and 50 mg/mL were prepared, respectively. The hamsters were administrated with a volume 1 ml of melatonin solution/kg of BW. For example: the body weight is 150 g, and the melatonin in ethanol solution was given to hamster 150 μL. I have added the point in our revised manuscript (Page 2, Lines 64-67).

 Q2. Time of daily melatonin dosage is not provided - was it in the evening, in accordance with the physiological melatonin secretion? This information is very important for the practical use of the results and planning further experiments, especially in humans.

Answer: Thanks for reviewer suggestion. I have added it in our revised manuscript  (Page 2, Line 68).

 3. In Material and Methods section it should be rather adipose/liver tissue sample/samples, not adipose/liver tissues.

Answer: Thanks for reviewer suggestion. I have modified them in Material and Methods section of our revised manuscript.

 4. The term "body mass' (BM) instead of "body weight' (BW) should be used throughout the manuscript.

Answer: Thanks for reviewer suggestion. I have modified them in our revised manuscript.

 5. In Table 2, the superscripts a, b, c are not described enough - which groups are compared as a, b, or c? The explanation is incomprehensible. The same problem is in Figures 1, 2, 4.

Answer: Thanks for reviewer suggestion. I have modified them in our revised manuscript (Table 2, and Figure 1, 2, 4, 5).